# *Kudoa septempunctata* Spores Cause Acute Gastroenteric Symptoms in Mouse and Musk Shrew Models as Evidenced In Vitro in Human Colon Cells

**DOI:** 10.3390/pathogens12050739

**Published:** 2023-05-20

**Authors:** Sung-Hee Hong, Ji-Young Kwon, Soon-Ok Lee, Hee-Il Lee, Sung-Jong Hong, Jung-Won Ju

**Affiliations:** 1Division of Vectors and Parasitic Diseases, Korea Disease Control and Prevention Agency, 187 Osongsaenmyeong2-ro, Osong-eup, Heungdeok-gu, Cheongju 28159, Republic of Korea; h22h22h22@korea.kr (S.-H.H.); kjiy31@korea.kr (J.-Y.K.); isak@korea.kr (H.-I.L.); 2College of Medicine, Chung-Ang University, Seoul 06974, Republic of Korea; amkey68@khu.ac.kr; 3Department of Medical Research Center for Bioreaction to Reactive Oxygen Species, Biomedical Science Institute, School of Medicine, Graduate School, Kyung Hee University, Seoul 02447, Republic of Korea; 4Convergence Research Center for Insect Vectors, Incheon National University, Incheon 22012, Republic of Korea

**Keywords:** *Kudoa septempunctata*, diarrhea, emesis, serotonin, suckling mouse, house shrew

## Abstract

*Kudoa septempunctata* is a myxosporean parasite that infects the trunk muscles of olive flounder (*Paralichthys olivaceus*) and has been reported to cause foodborne illnesses in humans. However, the molecular mechanisms underlying *K. septempunctata* spore toxicity remain largely unknown. In this study, the gastroenteropathy of *K. septempunctata* was examined in human colon adenocarcinoma cells as well as experimental mice inoculated with spores. We found that *K. septempunctata* decreased transepithelial resistance and disrupted epithelial tight junctions by deleting ZO-1 in Caco-2 monolayers. Additionally, serotonin (5-HT), an emetic neurotransmitter, was increased in *K. septempunctata*-inoculated cells. In vivo, *K. septempunctata* spores induced diarrhea in suckling mice (80% in ddY and 70% in ICR mice), with a minimum provocative dose of 2 × 10^5^ K. *septempunctata* spores. In house musk shrews, *K. septempunctata* induced emesis within 1 h and induced serotonin secretion in the intestinal epithelium. In conclusion, *K. septempunctata* may induce diarrhea and emesis by increasing intestinal permeability and serotonin secretion.

## 1. Introduction

*Kudoa septempunctata* (Myxosporea: Multivalvulida) is a myxosporean, which is primarily found in the trunk muscles of olive flounders (*Paralichthys olivaceus*). It has been found to occasionally cause transient diarrhea and vomiting in humans in Japan [1,2,3,4,5]. *K*. *septempunctata* spores have six or seven shell valves and polar capsules. *K*. *septempunctata* can be genetically classified into ST1, ST2, and ST3 groups; both ST1 and ST2 genotypes are typically found in Japan, whilst ST3 is dominant in the Republic of Korea [2]. The initial case report described *K*. *septempunctata* as the causative agent of food poisoning in Japan, and it has previously been identified as a neglected foodborne pathogen that can cause food poisoning [1,4,5,6,7,8]. However, the pathogenic mechanism of this myxosporean is not yet entirely understood.

The pathogenicity of *K*. *septempunctata* has previously been investigated using experimental mouse models, including house musk shrews (*Suncus murinus*) and suckling mice. The symptoms, including diarrhea and emesis, were transient. The lag phase was found to be only 1–20 h after feeding with olive founder fillets containing *K*. *septempunctata* spores [5]. The pathogenicity of *K*. *septempunctata* was first observed in experiments using suckling ddY mice. Other past studies have reported no pathological changes in adult BALB/c and ddY suckling mice that had been fed *K*. *septempunctata* spores of genotype ST3 [9,10,11].

The pathogenicity of *K*. *septempunctata*, as determined through in vitro assays using Kudoa spores, remains unclear [12,13,14]. The toxicity of *K*. *septempunctata* has been verified by measuring the tight junction barrier integrity of Caco-2 cells [12]. The sporoplasm of *K*. *septempunctata* has been shown to quickly invade the cytoplasm of Caco-2 cells. Invasion results in the rapid decrease in transepithelial electrical resistance (TEER) [15]. Another previous study reported the immune-promoting effects of *K*. *septempunctata* in HT29 cells [16]. Conversely, another study reported that *K*. *septempunctata* spores did not cause inflammation in Caco-2 cells [13]. Furthermore, Yamamoto et al. insisted that *K*. *septempunctata* damaged the human intestinal epithelium to increase the production of IL-8 and serotonin (5-HT), which likely resulted in emesis associated with *K*. *septempunctata* invasion [17].

To identify the pathogenic mechanism of *K*. *septempunctata*, here, we analyzed the permeability of human intestinal epithelial cells inoculated with *K*. *septempunctata* alongside the pathogenicity of *K*. *septempunctata* in suckling mice and house musk shrews.

## 2. Materials and Methods

### 2.1. Experimental Animals and Ethics

House musk shrews (*Suncus murinus*, 8 weeks old) were purchased from JLA Inc. (Naritashi, Japan). The animal protocol used in this study was reviewed and approved based on ethical procedures and scientific care by the Korea Disease Control & Prevention Institutional Animal Care and Use Committee (KDCA-IACUC; Permit Number KDCA-02-19-2A). House musk shrew care and protocols were carried out in accordance with the Korean Disease Control guidelines for the Care and Use of Laboratory Animals. Additionally, specific pathogen-free, pregnant ddY mice were purchased from Keari Co., Ltd. (Wakayama, Japan) and bred at the Laboratory Animal Facility at Chung-Ang University (Seoul, Republic of Korea). Meanwhile, ICR suckling mice were purchased from YoungBio Ltd. (Seongnam-si, Republic of Korea). All animal experiments were performed in accordance with the Chung-Ang University Guide for the Care and Use of Laboratory Animals (Permit Number:2020-00015).

### 2.2. Kudoa septempunctata Collection

Olive flounders were purchased from fishery farms on Jeju Island, with the flounders being subsequently examined for *K*. *septempunctata* spores using rapid diagnostic kits (Kudoa Rapido A; Solpotoo Co., Ltd., Jeju, Korea) by following the manufacturer’s instructions. *K*. *septempunctata* -infected olive flounder muscles (>10^6^ spores/g) were then minced in phosphate-buffered saline (PBS) before being centrifuged at 2000× *g* for 10 min. The pellet was then resuspended in PBS and placed in a 15–30% Percoll gradient cushion (Sigma, St. Louis, MO, USA) within a 15 mL conical tube. After centrifugation at 1500× *g* for 30 min at 20 °C, the resulting pellets of Kudoa spores were washed by resuspension in 50 mL of PBS and then spun at 1500× *g* for 30 min at 20 °C. Finally, spores were collected in 1–2 mL of PBS and subsequently counted using a hematocytometer [3]. The spores were then stored at –70 °C until further use. Additionally, the spore suspensions for the inactivation were heated at 95 °C for 10 min to inactivate the spores [5].

### 2.3. Correlative Three-Dimensional (3-D) Quantitative Phase Imaging (QPI)

For 3-D QPI, *K*. *septempunctata* was sub-cultured in a microscopic dish (Tomodish, Tomocube Inc., Daejeon, Republic of Korea) of #1.5H thickness with a 50 mm diameter glass bottom. The 3-D QPI of live *K. septempunctata* was obtained using commercial holotomography (HT-2H, Tomocube Inc., Daejeon, Republic of Korea), which is based on Mach–Zehnder interferometry equipped with a digital micromirror device (DMD). A coherent monochromatic laser (λ = 532 nm) was divided into two paths, a reference and a sample beam, using a 2 × 2 single-mode fiber coupler. A 3-D refractive index (RI) tomogram was then reconstructed using multiple 2-D holographic images acquired under 49 illumination conditions, normal incidence, and 48 azimuthally symmetric directions with a polar angle of 64.5. The DMD was used to control the angle of the illuminating beam impinging on the sample [18]. The diffracted beams from the sample were subsequently collected using a high numerical aperture (NA) objective lens (NA = 1.2, UPLSAPO 60XW, Olympus, Tokyo, Japan). The off-axis hologram was also recorded using a CMOS image sensor (FL3-U3-13Y3MC; FLIR Systems). A total of 33 2-D sections within a 5 μm range were acquired by moving the focus along the z-axis with a step size of 156 nm, immediately after acquiring a 3-D QPI image. Deconvolution of the reconstructed 3-D fluorescence images was performed using commercial software (AutoQuant X3, Media Cybernetics, Rockville, MD, USA). Additionally, the visualization of 3-D RI maps was carried out using commercial software (TomoStudioTM, Tomocube Inc., Daejeon, Republic of Korea).

### 2.4. TEER Assay for Spore Activity

Human intestinal cell line Caco-2 cells (ATCC, HTB37; Manassas, VA, USA) were used to assess the activity of *K*. *septempunctata* spores collected from olive flounder muscles. For this, the Caco-2 cells were maintained in Dulbecco’s modified Eagle medium (Gibco, St. Louis, MO, USA) supplemented with 10% FBS, 100 U/mL penicillin, and 100 μg/mL streptomycin (Gibco, St. Louis, MO, USA) at 37 °C in a 5% CO_2_ incubator. For these assays, 24-well plates with collagen-coated insert membranes (Corning, New York, NY, USA) were used. Caco-2 cells were seeded at 2 × 10^5^ cells/well on Biocoat cell culture inserts (pore size of 0.4 μm). TEER was then measured using a Millicell ERS-2 (Millipore, Billerica, MA, USA) to determine the state of differentiation and cell monolayer integrity, with the inserts with TEER > 1000 Ω·cm^2^ being subjected to downstream experiments. Purified *K*. *septempunctata* spores (5 × 10^5^ spores/well) were then added to cell culture inserts, with TEER being measured every hour. Heat-inactivated spores (equivalent to 5 × 10^5^ spores/well) were also inoculated as a control in cell culture inserts, with TEER again being measured.

### 2.5. Immunofluorescence Assay on Zonula occludens-1 (ZO-1) in Cell-Line Cells

The Caco-2 or HCA-7 cell monolayers treated with *K*. *septempunctata* live spores (5 × 10^5^ spores/well) or the heat-inactivated spores (equivalent to 5 × 10^5^ spores/well) were incubated for 4 h at 37 °C. The treated cells were fixed in 4% formaldehyde solution (Sigma–Aldrich, St. Louis, MO, USA) overnight at 4 °C and then washed with PBS. After permeabilization with 0.1% Triton X-100 (Sigma–Aldrich, St. Louis, MO, USA) for 5 min, the cells were blocked with 1% BSA for 30 min before being incubated with Zonula occludens-1-FITC conjugated antibody (1:200, Invitrogen, Waltham, MA, USA) overnight at 4 °C in humidified chambers. The labelled cells were then washed in PBS and observed under a fluorescence microscope (Olympus, Tokyo, Japan).

### 2.6. Assay on the Serotonin Production of Caco-2 Cells

After TEER measurement, Caco-2 cell culture media were collected [17], whilst the concentration of serotonin (5-hydroxytryptamine, 5-HT) was measured using an ELISA kit (IBL, Minneapolis, MN, USA), in accordance with the manufacturer’s instructions. 

### 2.7. Suckling Mouse Provocation for Diarrheal Response

All suckling mice were handled throughout the experiment, received from a delivery, stabilized for 1 h, administered the spores, and then observed for diarrhea in a walk-in-incubator that was maintained at 27 °C and 50% humidity. Suckling mice (4–5 days old, ddY strain) were separated from their mothers 2 h prior to administration of the spores and randomly divided into four groups (*n* = 10): *K*. *septempunctata*, 2 × 10^6^ and 2 × 10^5^ spores; heat-inactivated *K*. *septempunctata*, 2 × 10^6^ spores; and PBS only, as a negative control. 

ICR suckling mice (4–5 days old) were used to explore whether *K*. *septempunctata* spores could provoke diarrhea in this mouse strain. ICR suckling mice were directly delivered from the animal supplier early in the morning on the day of the experiment before being randomly divided into six groups (*n* = 5 or 10) as follows: *K*. *septempunctata* 2 × 10^6^, 2 × 10^5^, 2 × 10^4^, 2 × 10^3^, 2 × 10^2^ spores; heat-inactivated *K*. *septempunctata*, 2 × 10^6^ spores; and PBS only, as a negative control. 

The spores were then suspended in 100 μL of PBS before being administered directly into the stomach of suckling mice using a 1 mL syringe and needle capped with a polyethylene capillary tube (Appendix A). All suckling mice in this experiment were laid individually on a paper towel in a plastic weigh boat under incandescent lamps in a walk-in-incubator at a temperature of 27 °C and 50% humidity. The ambient temperature of the suckling mice was maintained at 31–35 °C in this study (Appendix A). We also observed and recorded mouse movements, defecation, fecal form, and diarrhea over a time course of 4 h. 

### 2.8. Experimental Design for the House Musk Shrews to Determine Emetic Response to K. septempunctata

House musk shrews (8 weeks old, *n* = 25) were randomly divided into four groups: Heat-inactivated *K*. *septempunctata* 1 × 10^8^ spores (*n* = 8); *K*. *septempunctata* 1 × 10^7^ spores (*n* = 8); *K*. *septempunctata* 1 × 10^8^ spores (*n* = 7); and PBS alone as a negative control. The spores were suspended in 500 μL PBS and orally administered using a ball-pointed needle at room temperature (18–20 °C). We then observed the shrews for 40 min and recorded their responses, including emesis and nausea.

### 2.9. Immunohistochemical Localization of Serotonin in the Intestinal Tissues

The small and large intestines of house musk shrews were both resected 1–2 h after oral administration of the spores and fixed using the Swiss roll method [19]. Paraffin blocks were sectioned at 5–6 μm thickness and deparaffinized. For immunohistochemical staining, ribbons were rehydrated, with endogenous peroxidases being quenched with 3% H_2_O_2_ in methanol for 30 min in the dark. After washing with PBS for 20 min, the ribbons were blocked with 1% bovine serum albumin (Sigma–Aldrich, St. Louis, MO, USA) for 20 min. The ribbons were then incubated with primary antibodies at the following dilutions in blocking buffer: Mouse anti-serotonin (1:250, LSBio, Seattle, WA, USA), anti-c-fos (1:100, Abcam, Cambridge, UK), anti-mast cell tryptase (1:200, Abcam), anti-chromogranin A (1:400, Abcam), and anti-CaMKI (1:50, Abcam). After washing in PBS, the ribbons were incubated with the secondary antibody, horseradish peroxidase-conjugated anti-mouse IgG (1:500, Cell Signaling, Danvers, MA, USA). After washing, the color was developed using a substrate, DAB (3,3′-diaminobenzidine tetrahydrochloride), before being counter-stained with hematoxylin.

### 2.10. Statistical Analysis 

Student’s *t*-tests were used to evaluate the statistical significance of the differences between individual experimental groups. Differences were considered statistically significant at *p* < 0.05.

## 3. Results

### 3.1. Kudoa septempunctata Sporoplasm Altered the Transepithelial Electrical Resistance of the Caco-2 Cell Monolayer

A cell monolayer permeability assay was performed to examine whether *K*. *septempunctata* damaged the Caco-2 cell junctions. When added to the Caco-2 cell monolayers, the live *K*. *septempunctata* spores released sporoplasm, whilst the heat-inactivated *K*. *septempunctata* spores did not (Figure 1A). In addition, it was observed as a result of holotomographic microscopy that the sporoplasm of *K*. *septempunctata* was released within 30 min after exposure to 37 °C (the temperature that the spores are exposed to in the stomach and intestines during human infection) but were not released at room temperature (18–20 °C) (Figure 1B). TEER across the Caco-2 cell monolayer was measured as an index of permeability change. The live *K*. *septempunctata* spores substantially decreased TEER by 70%, 1 h after inoculation, indicating an increase in the permeability of the Caco-2 cell monolayer (Figure 1C). These results indicated that *K*. *septempunctata* could interfere with the maintenance of TEER of Caco-2 cell monolayers through releasing sporoplasm at 37 °C. 

### 3.2. K. septempunctata Disturbed Tight Junction ZO-1 Protein Arrangement

ZO-1 protein decreased within the monolayer of intestinal epithelial cells (Caco-2 and HCA-7) treated with live *K*. *septempunctata* spores after 4 h (Figure 2). There was no difference observed in the expression of ZO-1 protein in intestinal epithelial cells that had been treated with heated *K*. *septempunctata* spores. The results subsequently revealed that *K*. *septempunctata* sporoplasm was invasive and increased the permeability of the intestinal epithelium by disturbing the ZO-1 protein arrangement in the tight junction.

### 3.3. Diarrheal Responses of ddY and ICR Suckling Mice to K. septempunctata

To confirm whether *K*. *septempunctata* caused diarrhea, spores were orally administered to ddY suckling mice. The *K*. *septempunctata* 1 × 10^5^ spore group produced diarrhea 1.3 times in 40% of the mice (Table 1) for 119.0 ± 32.8 min. In contrast, the 2 × 10^6^ *K. septempunctata* group spores provoked diarrhea 3.3 times in 80% of the ddY suckling mice (*p* < 0.001) for 115.0 ± 23.9 min. The duration of diarrhea was similar between the two groups, although the frequency of diarrhea was increased in the high-dose group (2 × 10^6^).

In ICR suckling mice, the *K*. *septempunctata* 2 × 10^6^ spore group provoked diarrhea 4.3 times in 100% of the mice, whilst the 2 × 10^5^ spore group resulted in diarrhea 2.4 times in 80% of the mice (*p* < 0.001) (Table 2). In the 2 × 10^4^ and 2 × 10^3^ spore groups, one of the five mice had diarrhea once. Meanwhile, in the 2 × 10^2^ spore group, the mice did not exhibit diarrhea or acute gastrointestinal symptoms (Table 2). These results indicated that high doses of *K*. *septempunctata* spores produced diarrhea one to four times in ddY and ICR suckling mice at an ambient temperature of 31–35 °C.

### 3.4. Emetic Response to K. septempunctata in House Musk Shrews 

Emetic responses were induced in three of the four house musk shrews that were administered the *K*. *septempunctata* 1 × 10^8^ spores (Table 3). It took 30.0 ± 5.8 min to induce emesis, with an average of 4.0 ± 1.2 emetic reactions. In the 1 × 10^7^ spore group, emesis was induced in two of four house musk shrews with 3.0 ± 1.0 emetic reactions in 23.5 ± 2.5 min. However, no emetic reactions were observed in the heat-inactivated *K*. *septempunctata* group.

### 3.5. Serotonin (5-HT) Produced In Vitro and In Vivo by K. septempunctata

An additional experiment was conducted to determine whether serotonin, which plays an important role in inducing emesis, was secreted from intestinal epithelial cells, Caco-2 cells, by *K*. *septempunctata* (Figure 3). The concentration of serotonin secretions increased after 4 h of treatment with *K*. *septempunctata* in Caco-2 cells compared to the control group (*p* < 0.01). To confirm the mechanism of serotonin-induced vomiting, the induction of serotonin secretion was confirmed in the intestinal tissues of house musk shrews that had been orally administered *K*. *septempunctata*. Furthermore, serotonin secretion was induced more both in the small and large intestines of the house musk shrews than in the control and heat-inactivated *K*. *septempunctata* groups (Figure 4).

## 4. Discussion

The food-borne parasite, *K*. *septempunctata*, can parasitize humans accidentally when raw or undercooked parasitized fish are consumed. Additionally, altered intestinal permeability plays a significant role in the worsening of the clinical manifestations in patients where hostile reactions to food have been demonstrated [20,21,22]. The habit of eating raw fish therefore poses a high risk to the integrity of the intestinal mucosa [5,7,23]. The sporoplasm released from *K*. *septempunctata* spores contributes to gastrointestinal symptoms and occasionally results in concurrent inflammatory symptoms [6,16,24,25].

Intestinal barrier dysfunction associated with gastrointestinal symptoms influences both the sensitization and effector phases of toxicity. Caco-2 cells have been extensively used as an in vitro model for the evaluation of intestinal permeability considering their structural and functional similarity to the mature intestinal epithelium [26,27,28]. The *K*. *septempunctata* spores were found to have rapidly decreased TEER (within 1 h), thus indicating an increase in the permeability of the Caco-2 cell monolayer. The early decrease in TEER observed in Caco-2 monolayers could be explained by the release of sporoplasm from *K*. *septempunctata*. In this study, the release of sporoplasm from *K*. *septempunctata* spores was observed on the surface of Caco-2 cells within 1 h of inoculation. Several myxosporean parasites have previously been observed to release sporoplasm into the host intestine [12,29]. The sporoplasm of *K*. *septempunctata* quickly invaded Caco-2 cells, reaching the basolateral side of Caco-2 cells within 30 min of infection. This rapid invasion corresponded to a reduction in TEER alongside a short lag phase before the emergence of clinical symptoms. In addition, the release of sporoplasm at 37 °C (human body temperature) could be suggested as a reason for the acute diarrhea and emesis resulting from *K*. *septempunctata* infection. Our results confirmed that the *K*. *septempunctata* sporoplasm invasion could increase the permeability of the human intestinal epithelium and cause acute diarrhea. 

Diarrheal diseases caused by enteric infections can result in impaired epithelial barrier function and the dysregulation of fluid and ion transport procedures [30,31,32,33]. Several enteric pathogens have established specific strategies to change or disrupt this structure as part of pathogenesis, resulting in either pathogen invasion or consequences, such as diarrhea [30,34]. Previous reports have shown that diarrhea-inducing parasites were associated with an increased permeability of the intestinal epithelium, alongside decreased tight junction (TJ) proteins, such as occludin and ZO-1 [28,35,36]. The TJ complex is composed of transmembrane proteins, including occludin, junctional adhesion proteins, and members of the claudin family. ZO-1 interacts with the intracellular protein complex that attaches to structural elements of the cytoskeleton [37]. ZO-1 is a key player in the interaction of occludin with other TJ junction proteins. Our results provide evidence that the release of *K*. *septempunctata* sporoplasm interfered with its interaction with ZO-1 and other TJ proteins, thus inducing diarrhea. 

There have been several previous reports investigating whether *K*. *septempunctata* spores were able to provoke diarrhea in suckling mice. For example, Jang et al. reported that *K*. *septempunctata* did not infect the gastrointestinal tract or induce diarrhea in ddY suckling mice [9]. Meanwhile, Kawai et al. suggested that *K*. *septempunctata* induced watery stools and elevated fluid accumulation in suckling mice [5]. In the present study, *K*. *septempunctata* induced diarrhea in ICR and ddY suckling mice. An explanation for these varying experimental results for suckling rats as a result of Kudoa infection could be due to the experimental temperature conditions. The digestive physiology of suckling mice was observed to decrease at a room temperature of 21 °C, although the body temperature of suckling mice was maintained at normal levels at 31–35 °C, whilst digestive physiology continued to function normally in response to *K*. *septempunctata* infection. In addition, a minimal-dose assay for diarrhea-inducing *K*. *septempunctata* in suckling ICR mice was conducted. The 2 × 10^6^ and 2 × 10^5^ spore groups each induced diarrhea in 100% and 80% of ICR suckling mice, respectively, whilst the 2 × 10^4^ and 2 × 10^3^ spore groups induced diarrhea in only 20% of ICR suckling mice. Additionally, Kawai et al. found that 1 × 10^5^ *K. septempunctata* spores induced diarrhea in ddY suckling mice. The results for the minimum dose required to induce diarrhea in suckling mice provide evidence for the estimated intake threshold of *K*. *septempunctata* for the occurrence of symptoms in humans being 7.2 × 10^7^ spores/g of fish filet [24].

The pathophysiological basis of the derivation of emesis by *K*. *septempunctata* remains poorly understood. The luminal enterochromaffin (EC) cells “taste” and “sense” the luminal contents and can release mediators such as serotonin to activate ENS, as well as extrinsic vagal afferents to the brain. The stimulation of the extrinsic vagal and spinal afferent fibers by serotonin slows gastric emptying, pancreatic secretion, satiation, pain, and discomfort, alongside nausea and emesis [38]. Serotonin is also present in the secretory granules of EC cells, which are highly abundant in the duodenum and comprise the single largest enteroendocrine cell population [39,40]. The main depository of serotonin in the body is the mucosa of the gastrointestinal tract [41], although the details of gut serotonin remain unclear [42]. Rotavirus has previously been shown to stimulate human EC cells in the gut, causing serotonin release, which activates vagal afferents and the brain stem emesis center, a reaction cascade associated with emesis [43]. It has been previously reported that the oral administration of *K*. *septempunctata* to house musk shrews induced serotonin production in the intestines [44]. In this study, serotonin increased in response to the *K*. *septempunctata* cascade in Caco-2 cells and in the small and large intestinal tissues of house musk shrews. Serotonin is a physiological mediator that stimulates the emesis center in the brain. It has been suggested that serotonin produced in the intestine may trigger disease-associated emesis [45,46,47]. Additionally, serotonin released from EC cells acts as a pro- and anti-inflammatory molecule, which is strongly supported by the evidence. IL-8 and serotonin have both been reported to cause inflammation and emesis associated with *K*. *septempunctata* invasion via AP-1 upregulation and the induction of the MAPK/NF-kappa B pathway [17]. In the current study, serotonin induced by *K*. *septempunctata* spores played a role in emesis in house musk shrews.

Based on the above results, it is proposed that the *K*. *septempunctata* spores released the sporoplasm at 37 °C decreased TEER in the intestinal epithelium, which consequently transuded exudate into the intestinal lumen and caused watery diarrhea in suckling mice. The spores also induced serotonin production in the intestine, which in turn mediated emesis in house musk shrews.

## Figures and Tables

**Figure 1 pathogens-12-00739-f001:**
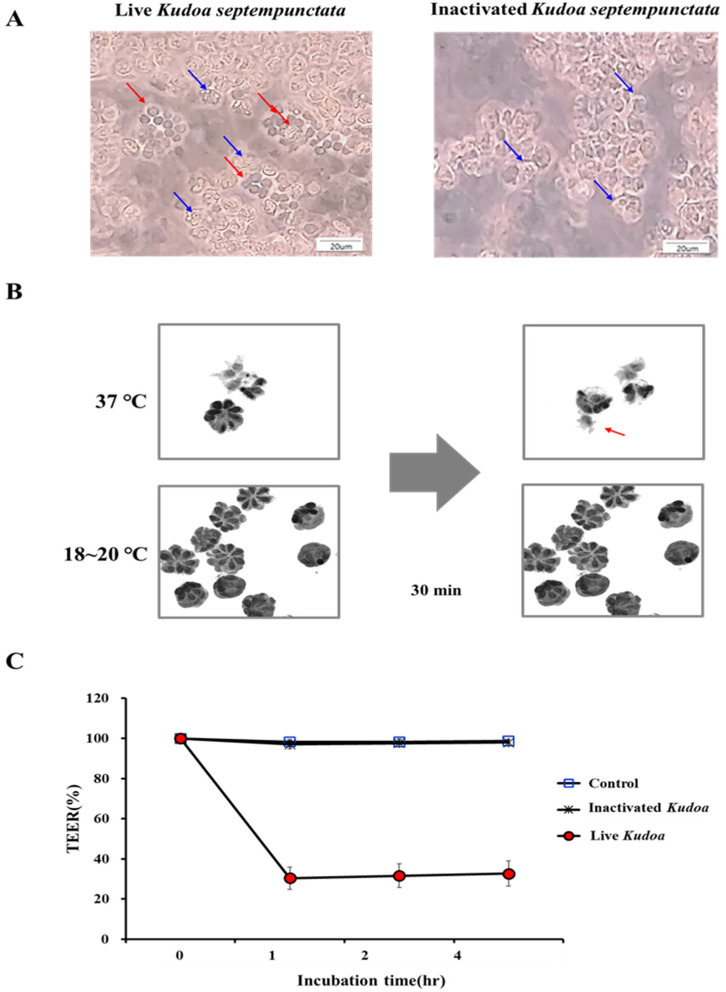
Release of sporoplasm from *Kudoa septempunctata* spores and the alteration of transepithelial electrical resistance (TEER) in Caco-2 cell monolayers. (**A**) The *K. septempunctata* spores (blue arrow) were added to the Caco-2 cell monolayer and incubated at 37 °C for 1 h. Sporoplasm released is indicated (red arrows). (**B**) A sporoplasm releasing (red arrow) from a *K*. *septempunctata* spore was observed under a holotomographic microscope 30 min after incubation at 37 °C. (**C**) TEER decrease in the Caco-2 cell monolayer by live *K*. *septempunctata* spores. Values are means ± standard deviation; measurements in triplicate.

**Figure 2 pathogens-12-00739-f002:**
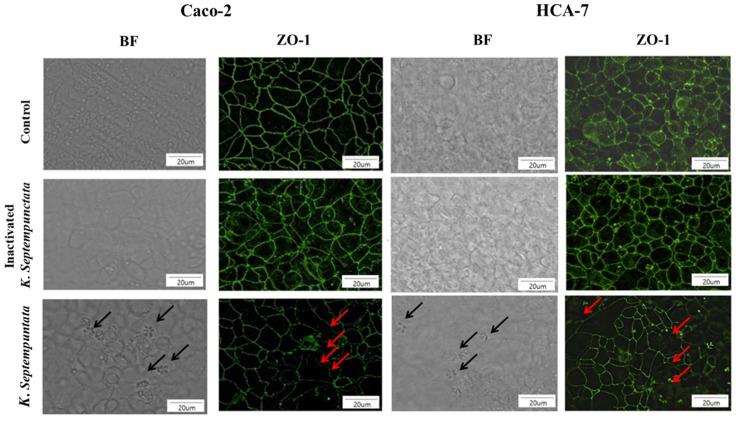
Zonula occuludens-1 (ZO-1) localized by immunofluorescent staining. ZO-1 expression revealed altered matrices in Caco-2 and HCA-7 cells incubated with *K. septempunctata* for 4 h. Black arrows indicate *K*. *septempunctata* sporoplasm on cells. Red arrows indicate disturbing the ZO-1 proteins. BF, bright field.

**Figure 3 pathogens-12-00739-f003:**
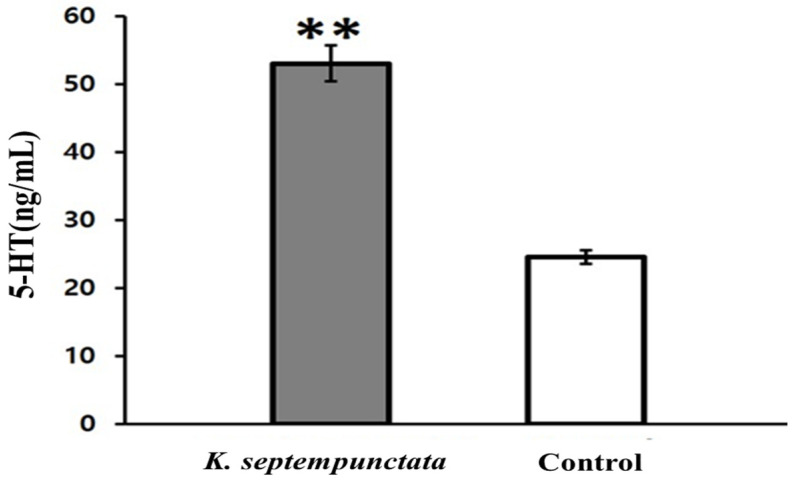
Serotonin (5-HT) produced and secreted from Caco-2 cells. *K*. *septempunctata* spores were added to the culture medium and cultured for 4 h. Serotonin levels were determined in the culture supernatants. Data represent the mean ± standard deviation (SD) of three independent replicates. ** Statistically significant difference between treated and untreated cells (*p* < 0.01).

**Figure 4 pathogens-12-00739-f004:**
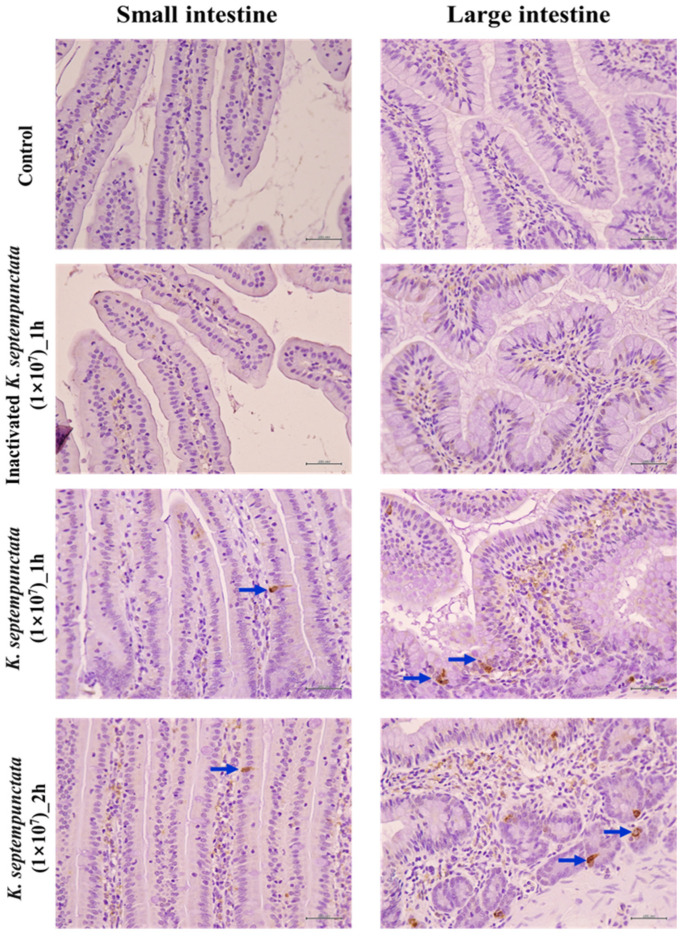
Localization of serotonin in the epithelia of the small and large intestines of house musk shrews administered 1 × 10^7^ *K. septempunctata* spores/mouse. Blue arrows indicate serotonin-producing cells; scale bar = 200 nm.

**Table 1 pathogens-12-00739-t001:** Diarrheal response of ddY suckling mice provoked by *Kudoa septempunctata* spores.

*K. septempunctata* Spores Administered.	No. of Diarrhea/Mouse Tested	No. of Diarrheal Incidents	Duration of Diarrhea (min)
Live 2 × 10^6^	8/10 ***	3.3	115.0 ± 23.9
Live 1 × 10^5^	4/10	1.3	119.0 ± 32.8
Heat-inactivated 2 × 10^6^	0/10	_	_
PBS	1/10	1	254

*** Differences between the treated and untreated groups; statistical significance, *p* < 0.001. Value represents the mean ± standard deviation.

**Table 2 pathogens-12-00739-t002:** Diarrheal response of ICR suckling mice to *Kudoa septempunctata* spores.

*K. septempunctata* Spores Administered	No. of Diarrhea/Mouse Tested	No. of Diarrheal Incidents	*p*-Value
Live 2 × 10^6^	10/10	4.3	0.0001 ***
Live 2 × 10^5^	8/10	2.4	0.0006 ***
Live 2 × 10^4^	1/5	0.4	0.555
Live 2 × 10^3^	1/5	0.4	0.555
Live 2 × 10^2^	0/5	0	-
Heat-inactivated 2 × 10^6^	0/5	0.1	-
PBS	1/10	0.1	-

*** Differences between the treated and untreated groups; statistical significance, *p* < 0.001.

**Table 3 pathogens-12-00739-t003:** *Kudoa septempunctata* spore-induced emetic responses in house musk shrews.

*K. septempunctata* Spores Administered	No. of Musk Shrews (Emesis/Tested)	No. of Emetic Events	Duration of Emesis (min)
Live 1 × 10^7^	6/7 ***	4.0 ± 1.2	30.0 ± 5.8
Live 1 × 10^6^	4/8	3.0 ± 1.0	23.5 ± 2.5
Heat-inactivated 1 × 10^7^	0/8	0	0
PBS	0/2	0	0

*** Differences between the treated and untreated groups; statistical significance, *p* < 0.001. Values represent the mean ± standard deviation.

## Data Availability

Not applicable.

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
