# Peer review of "Kudoa septempunctata Spores Cause Acute Gastroenteric Symptoms in Mouse and Musk Shrew Models as Evidenced In Vitro in Human Colon Cells"

_pathogens, 2023, doi:10.3390/pathogens12050739_

Round 1

Reviewer 1 Report

My main concern with the study is relative to the viability of the K.septempunctata spores used. These are collected from infected fish muscle and frozen at -70C without cryoprotectants or vitrification protocols. However, when used in the experiments in vitro, the authors observations seem to imply that they are alive and that the sporoplasms germinate and invade the cells as if they were biologically viable. This is unlikely and would be indeed quite extraordinary. In addition, according to the knowledge available for myxozoan life cycles, normal germination of these spores should happen in marine annelids (quite different physical-chemical conditions to in vitro-cultured human cells) and involve extrusion of polar filaments. The images by 3D QPI, while nice, do not clearly show these processes. The response to the spores and the differences with the control and heat-inactivated spores appear convincing, but the authors should conciliate the results with other processes different to biological "germination" of the spores and "invasion or "parasitizing" of the sporoplasms, in the discussion section. Chemical toxicity is not to be disregarded

Minor observations: Please note exponent numbers properly throughout the MS. Use italics in latin names and expressions (e.g., in vitro). Move Fig 1 to results section   

The MS is understandable and only requires minor English editing.

Author Response

Response to Reviewer 1 Comments

Comments and Suggestions for Authors

Point 1: My main concern with the study is relative to the viability of the K.septempunctata spores used. These are collected from infected fish muscle and frozen at -70C without cryoprotectants or vitrification protocols. However, when used in the experiments in vitro, the authors observations seem to imply that they are alive and that the sporoplasms germinate and invade the cells as if they were biologically viable. This is unlikely and would be indeed quite extraordinary. In addition, according to the knowledge available for myxozoan life cycles, normal germination of these spores should happen in marine annelids (quite different physical-chemical conditions to in vitro-cultured human cells) and involve extrusion of polar filaments. The images by 3D QPI, while nice, do not clearly show these processes. The response to the spores and the differences with the control and heat-inactivated spores appear convincing, but the authors should conciliate the results with other processes different to biological "germination" of the spores and "invasion or "parasitizing" of the sporoplasms, in the discussion section. Chemical toxicity is not to be disregarded.

Response 1: I appreciate your good comments.

Papers on cryopreservation of K.septempunctata have already been published,

(Cryopreservation of Kudoa septempunctata sporoplasm using commercial freezing media_ Parasitol Res. 2017 Jan;116(1):425-427)

As a result of storing the K.septempunctata cryopreservation in a composition that preserves cells (cryopreservation solution : FBS+10%DMSO), it was confirmed that sporoplasms was secreted from the K.septempunctata. In addition, it was confirmed that sporoplasms was not secreted when the K.septempunctata was cryopreserved at -70 degrees (in PBS) without cryopreservation solution.

We also performed TEER assay by treating K.septempunctata lysate to confirm whether it was caused by chemical toxicity, but no change occurred in intestinal epithelial cells.

Point 2:  Please note exponent numbers properly throughout the MS.

Response 2: All exponent numbers have been corrected.

Point 3: Use italics in latin names and expressions (e.g., in vitro).

Response 3: Italics are used for Latin names and expressions.

Point 4: Move Fig 1 to results section.   

Response 4: Moved Figure 1 to Results.

Point 5: Unclear: there is np insert in Fig A. The term “brood” also results unclear. (Line 203)

Response 5: Removed “Insert: A sporoplasm brood”.

Point 6: Abdominal pain is not mentioned elsewhere in the results section. How was this measured? If methods to record abdominal pain are not described and results are not given, this should be omitted. 

Response 6: Removed “abdominal pain”. (Line 163)

Point 7: Delete “and vitality”

Response 7: Removed “and vitality” (Line 108)

Point 8: Correct “heated spores”

Response 8: Corrected with heated inactivated spores. (Line 122)

Reviewer 2 Report

To identify the pathogenic mechanism of K. septempunctata, this study analyzed the permeability of human intestinal epithelial cells inoculated with K. septempunctata alongside the pathogenicity of K. septempunctata in suckling mice and house musk shrews. The experimental writing is relatively standardized, and there are several problems as follows:

 1. The title needs to be revised. The main purpose of this study is to reveal the generation of the molecular mechanisms underlying K. septempunctata spore toxicity, rather than simply proving that it can produce symptoms in suckling mice and house musk shrews.

2. Correction all the expressing the number of all spores in the text. Numbers of the order of magnitude should be superscripted

3. Line 39, Suncus murinus should be italic.

4. Line85-86, Reference should be provided for the method of inactivating spores

5. Figure 1 should provide a scale bar

6. The statistical analysis indicates that the difference is p<0.05, but there are many p<0.001 in the table. Please unify standards in statistical methods, such as *** represent p<0.001.

7. Table 1, **Value represents mean ± standard deviation, where is **?

8. Line 244, table 3, Kudoa septempunctata spore-induced emetic responses in house musk shrews. Why only 2 individuals in PBS?

9. Figure 4. Kudoa should be Kudoa septempunctata

Author Response

Response to Reviewer 2 Comments

To identify the pathogenic mechanism of K. septempunctata, this study analyzed the permeability of human intestinal epithelial cells inoculated with K K. septempunctata alongside the pathogenicity of K. septempunctata in suckling mice and house musk shrews. The experimental writing is relatively standardized, and there are several problems as follows:

Point 1: The title needs to be revised. The main purpose of this study is to reveal the generation of the molecular mechanisms underlying K. septempunctata spore toxicity, rather than simply proving that it can produce symptoms in suckling mice and house musk shrews.

Response 1: I appreciate your nice comments on the subject.

Unfortunately, the title was decided after consultation with the corresponding authors, and we also tried to do a more detailed analysis (signal pathway) to identify the mechanism, but the analysis did not proceed that far. I hope you can understand the difficulty of editing the title.

Point 2: Correction all the expressing the number of all spores in the text. Numbers of the order of magnitude should be superscripted

Response 2: All of them have been corrected.

Point 3. Line 39, Suncus murinus should be italic.

Response 3: It has been corrected.

Point 4. Line85-86, Reference should be provided for the method of inactivating spores

Response 4: References have been added [5].

Point 5. Figure 1 should provide a scale bar

Response 5: A scale bar has been added to Figure 1.

Point 6. The statistical analysis indicates that the difference is p<0.05, but there are many p<0.001 in the table. Please unify standards in statistical methods, such as *** represent p<0.001.

Response 6: I unified standards in statistical methods, such as *** represent p<0.001.

(Table 1, Table 2, Table 3) (Line 226, 236, 245)

Point 7. Table 1, **Value represents mean ± standard deviation, where is **?

Response 7: Removed “**” (Line 226, 245)

Point 8. Line 244, table 3, Kudoa septempunctata spore-induced emetic responses in house musk shrews. Why only 2 individuals in PBS?

Response 8: It was confirmed that there was no reaction to PBS in a muskrat pre-experiment on Kudoa septempunctata. So, in this experiment, the PBS group was set to 2 animals and the experiment was conducted. The results of the preliminary experiment were not added to the results of the main experiment because they did not see the results of the same time period.

Point 9. Figure 4. Kudoa should be Kudoa septempunctata

Response 9: The content of the picture has been corrected and added.